# Identification of *N*-Acetyl-*S*-(3-Cyano-2-(Methylsulfanyl)Propyl-Cysteine as a Major Human Urine Metabolite from the Epithionitrile 1-Cyano-2,3-Epithiopropane, the Main Glucosinolate Hydrolysis Product from Cabbage

**DOI:** 10.3390/nu11040908

**Published:** 2019-04-23

**Authors:** Franziska S. Hanschen, Susanne Baldermann, Adrian Brobrowski, Andrea Maikath, Melanie Wiesner-Reinhold, Sascha Rohn, Monika Schreiner

**Affiliations:** 1Plant Quality and Food Security, Leibniz Institute of Vegetable and Ornamental Crops, Theodor-Echtermeyer-Weg 1, 14979 Grossbeeren, Germany; baldermann@igzev.de (S.B.); Adrian.Bobrowski@gmx.de (A.B.); Maikath.Andrea@igzev.de (A.M.); wiesner@igzev.de (M.W.-R.); schreiner@igzev.de (M.S.); 2Hamburg School of Food Science, Institute of Food Chemistry, University of Hamburg, Grindelallee 117, 20146 Hamburg, Germany; rohn@chemie.uni-hamburg.de

**Keywords:** glucosinolate, epithionitrile, *Brassica*, metabolism, mercapturic acid

## Abstract

*Brassica* vegetables such as cabbage or pak choi contain alkenyl glucosinolates which can release epithionitriles and to a lesser degree isothiocyanates upon enzymatic hydrolysis. Here, for the first time, the metabolism of an epithionitrile was investigated in humans, namely 1-cyano-2,3-epithiopropane (CETP). After consumption of *Brassica oleracea* var. *capitata* f. *alba* and *Brassica carinata* sprouts, the main urinary metabolite of CETP was identified as *N*-acetyl-*S*-(3-cyano-2-(methylsulfanyl)propyl-cysteine using an UHPLC-ESI-QToF-MS approach and synthesis of the metabolite. This urinary epithionitrile metabolite is an *S*-methylated mercapturic acid. No other metabolites were detected. Then, in a preliminary pilot experiment the excretion kinetics of CETP were investigated in three volunteers. After consumption of a *B. carinata* sprout preparation containing 50.8 µmol of CETP, urinary *N*-acetyl-*S*-(3-cyano-2-(methylsulfanyl)propyl-cysteine concentrations were the highest three hours after consumption, ranging from 23.9 to 37.2 µM, and declined thereafter. Thus, epithionitriles are bioavailable compounds that are metabolized similarly to isothiocyanates by the mercapturic acid pathway. In the future, more epithionitrile metabolites should be identified and the pharmacokinetics of these important class of dietary compounds should be assessed in more detail.

## 1. Introduction

Epithionitriles are main glucosinolate hydrolysis products formed enzymatically in many *Brassica* species upon tissue disruption due to presence of the epithiospecifier protein (ESP) [1]. Among the more than 130 glucosinolates known so far [2], only a few have the terminal double bond that is a prerequisite for epithionitrile formation [3]. Nevertheless, their glucosinolate precursors allyl glucosinolate (sinigrin), 3-butenyl glucosinolate (gluconapin), 4-pentenyl glucosinolate (glucobrassicanapin), 2-hydroxy-3-butenyl glucosinolate (progoitrin and epiprogoitrin), and 2-hydroxy-4-pentenyl glucosinolate (gluconapoleiferin) are quite abundant in *Brassica oleracea* (e.g., white, red, and savoy cabbage) and *Brassica rapa* vegetables (e.g., pak choi), and release epithionitriles in quite high amounts [1,4,5,6]. In particular, allyl glucosinolate is widely distributed in the order Brassicales [7,8]. It is the most consumed aliphatic glucosinolate in Germany [9]. Therefore, its epithionitrile, 1-cyano-2,3-epithiopropane (CETP), can be considered as the most important epithionitrile in the human diet. The formation of CETP and other hydrolysis products from allyl glucosinolate is shown in Figure 1.

In the past decades, epithionitriles and other atypical products from glucosinolates have received intense attention in the pure plant biochemistry field [11,12,13,14]. However, in food and nutritional science, epithionitriles have been neglected and here glucosinolate research mainly focussed on isothiocyanates, another important class of glucosinolate hydrolysis products that are valued especially for their potential cancer-preventing effects [15]. As epithionitriles are so abundant in consumed vegetables they should not be ignored. With regard to their bioactivity, they showed nephrotoxic effects in animal experiments with rodents after oral gavage of 50–125 mg/kg body weight [16,17,18]. Conjugation with glutathione (GSH) is thought to be the initial step in epithionitrile metabolization but also toxification in rats [19], and renal cysteine *S*-conjugate *β*-lyases are likely candidates for the formation of the toxic rat metabolites [20]. In vitro, epithionitriles induced phase-II enzymes similarly to allyl isothiocyanate, with CETP being most potent among three epithionitriles tested [21]. On the other hand, CETP had no specific toxicity on cancer cells (in contrast to isothiocyanates), but induced cell death in primary mouse hepatocytes due to necrosis induction as well [4].

With regard to the metabolism of epithionitriles not much is known. Brocker et al. showed that ^35^S labelled CETP was mainly excreted via the urine and a mercapturic acid was tentatively identified as a urinary rat metabolite after derivatization (methylation) with diazomethane and gas chromatography-mass spectrometry (GC-MS) analysis [22]. In a similar approach, VanSteenhouse et al. tentatively identified *N*-acetyl-*S*-(4-cyano-2-thio-1-butyl)-cysteine as a urinary metabolite from 1-cyano-3,4-epithiobutane (CETB) [23]. Thus, conjugation with GSH and metabolism via the mercapturic acid pathway, being similar to the isothiocyanate metabolism [24], seems to be the main metabolic route of epithionitriles in rats [22,23].

With regard to human metabolism, no data are available thus far. Therefore, the aim of the present study was the identification of CETP metabolites in human urine and finally the investigation of the excretion kinetics of this compound by monitoring its concentrations in urine samples, as part of a human intervention pilot study.

## 2. Materials and Methods

### 2.1. Chemicals and Buffers

Allyl isothiocyanate (≥99%), *N*-acetyl-l-cysteine methyl ester (≥90%), and *N*-acetyl-l-cysteine (NAC, ≥99%) were purchased from Sigma-Aldrich Chemie GmbH (Steinheim, Germany). Acetic acid (100%), imidazole (p.A.), and formic acid were obtained from Carl Roth GmbH (Karlsruhe, Germany). Ethanol (absolute) was purchased from VWR International GmbH (Darmstadt, Germany). 3-Butenyl ITC (3But-ITC, ≥95%) was purchased from TCI Deutschland GmbH (Eschborn, Germany), and acetonitrile (LC-MS grade) was bought from Th. Geyer GmbH & Co. KG (Renningen, Germany). The epithionitrile 1-cyano-2,3-epithiopropane (CETP, ≥95%) was synthetized by Taros Chemicals GmbH Co. KG (Dortmund, Germany), and the epithionitrile 1-cyano-3,4-epithiobutane (≥99%, CETB) and the cysteine derivative *N*-Acetyl-*S*-(3-cyano-2-(methylsulfanyl)propyl-cysteine (≥98%) were synthesized by ASCA GmbH Angewandte Synthesechemie Adlershof (Berlin, Germany). Nuclear magnetic resonance (NMR) spectroscopy data of the cysteine derivative are provided in Appendix A. All solvents were of liquid chromatography-mass spectrometry (LC-MS) grade and water was of Milli-Q quality.

### 2.2. Initial Experiment for the Identification of Epithionitrile Metabolites

In a preliminary experiment, one of the authors ingested 5 g of white cabbage sprouts (*Brassica oleracea* var. *capitata* f. *alba* cv. Jetma RZ F1, Rijk Zwaan Welver GmbH, Welver, Germany) grown for 8 days on a wet fleece and put in a perlite/water mixture filled into an aluminum tray in a greenhouse. Prior to ingestion of the sprouts, the proband had a washout phase of 3 days, abstaining from Brassicales vegetables or other food containing glucosinolates or their hydrolysis products. After ingestion of the sprouts, total urine within the first 3 h, 3–6 h, and 6–24 h was sampled and stored at 4 °C for up to 18 h.

### 2.3. Preparation of Urine Samples and UHPLC-ESI-(Q)ToF-MS

Urine was centrifuged for 5 min in 50 mL tubes at 4 °C and 3400× *g*. Five milliliters of the supernatant were diluted with 5 mL of ethanol. Then, the sample again was centrifuged again to precipitate proteins as described above. Then, 500 µL of the supernatant were put into an Amicon Ultra 0.5 mL centrifugal filter (MWCO 3 kDa, Sigma-Aldrich Chemie GmbH, Steinheim, Germany) and at 10,000× *g* and 4 °C proteins were removed within 20 min. The filtrate was then stored at −80 °C until ultra-HPLC (UHPLC)-MS analysis. Samples were analyzed with an Agilent 1290 Infinity UHPLC coupled with an Agilent 6530 Accurate Mass Quadrupole Time-of-Flight (QToF) mass spectrometer (Agilent Technologies GmbH, Waldbronn, Germany). From a sample, 5 µL were injected onto a ZORBAX Extend-C18 column (2.1 × 50 mm, 1.8 μm, Agilent Technologies GmbH) and compounds were separated using a flow rate of 0.4 mL/min and a gradient of solvent A (0.01% formic acid in water) and B (0.01% formic acid in acetonitrile) as follows: 0–3 min 98% A, then a linear increase to 15% B within 7 min, to 40% B within 8 min, and to 100% B within 2 min, holding this percentage for 5 min. The oven temperature was set to 30 °C. Ionization was performed using an electrospray ionization (ESI) source in positive and negative ionization mode and MS spectra were collected at first in positive and negative ionization (acquisition rate 1 spectra/s, range *m*/*z* 90–1700, gas temperature 300 °C, capillary voltage 3500 kV, nebulizer gas flow 8 L/min at 35 psi, sheath gas temperature 320 °C, skimmer 65 V, fragmentor voltage, 100 V). The diode array detector (DAD) was set to 200–400 nm. Finally, the UHPLC-QToF-MS data were qualitatively analyzed using Agilent MassHunter Quantitative Analysis B.07.00. Possible metabolites were identified by searching the chromatograms for the exact masses of the [M + H]^+^ or [M − H]^−^ ion of potential metabolites formed by the mercapturic acid pathway (GSH conjugate, *S*-cysteinylglycine (Cys-Gly) conjugate, *S*-cysteinyl (Cys) conjugate, NAC conjugate) as well as methylated mercapturic acid pathway metabolites (single and double methylated). After the appearance of two possible newly formed metabolites with *m*/*z* 277.06751 and *m*/*z* 263.05186, MS/MS fragmentation experiments by QToF-MS were applied in the positive ionization mode for these compounds: The LC-MS settings were as described above, but for MS/MS the collision energies of 1, 3, 5, 10, 20, and 30 V were used to obtain mass spectra of the compounds.

### 2.4. Preparation of Potential Metabolites

Allyl isothiocyanate (1 mM) or CETP (1 mM) was either incubated with *N*-acetyl-l-cysteine methyl ester (5 mM) or NAC (5 mM) in an imidazole buffer at pH 8 in a total volume of 1 mL and room temperature (RT) for 1 h. 3-Butenyl isothiocyanate (1 mM) and CETB (1 mM) were similarly incubated in NAC (5 mM) and a total volume of 1 mL at pH 8 for 1 h. Then, samples were analyzed using the LC-MS method described above (positive ionization, MS-only and MS/MS with 5 V collision energy) and the retention times of the prepared conjugates were compared to those of the potential human CETP metabolites.

### 2.5. Plant Material For Human Pilot Intervention

For the human pilot intervention study, seeds of *Brassica carinata* (provided by the World Vegetable Center (AVRDC)) were sown on a wet fleece, and put in a perlite/water mixture filled into an aluminum tray. Sprouts were grown in a greenhouse for 8 days and watered when needed. Sprouts (27 g) were harvested and mixed with 27 mL of water and homogenized using a T25 digital ULTRA-TURRAX^®^ (IKA^®^-Werke GmbH & CO. KG, Staufen, Germany) leading to a beverage with a smoothie-like consistence. After 30 min of incubation, three replicates of 500 mg samples were analyzed for their glucosinolate hydrolysis product composition and content and the remaining beverage was immediately used for the intervention.

### 2.6. Human Intervention and Quantitation of the CETP-Metabolite in Urine Samples

In order to first obtain information about the excretion kinetics of the identified CETP-metabolite, a small human pilot study was conducted. Before the intervention, three authors volunteered for the self-trial and had a wash-out phase of one week, where they abstained from Brassicales vegetables or other food containing glucosinolates or their hydrolysis products. On the day of the intervention and directly before its start, each volunteer provided a urine sample (time zero). Then, each volunteer was supplied with 15.2 g of the prepared smoothie-like preparation (containing 7.6 g of *B. carinata* sprouts). Afterwards, subjects were allowed to drink a glass of water and to consume everything, with the exception of Brassicales vegetables. Urine was sampled after 3, 6, and 24 h (spot urine), and immediately centrifuged at 4 °C as described above and frozen at –80 °C until further clean-up. For the sample clean-up, the protocol described above was followed. Aliquots of the samples then were diluted 1:1 with water or 0.005 mM or 0.01 mM of an *N*-acetyl-*S*-(3-cyano-2-(methylsulfanyl)propyl-cysteine solution. Then, 5 µL of these dilutions were analyzed in positive ionization mode as described above and quantified via standard addition and based on the signal of the extracted ion mass of *m*/*z* 277.0675 and an extraction window of 20 ppm.

### 2.7. Analysis of Glucosinolate Hydrolysis Products in Plant Samples

Analysis of glucosinolate hydrolysis products from sprouts and in the beverage was performed as reported previously [1]. Each analysis was performed in triplicate.

## 3. Results

### 3.1. Epithionitriles and Other Glucosinolate Hydrolysis Products in Plant Material for Human Consumption

Homogenized white cabbage sprouts contained mainly CETP (4.26 ± 0.36 µmol/g fresh weight (FW)), but also small amounts of CETB (0.013 ± 0.005 µmol/g FW), 1-cyano-2-hydroxy-3,4-epithionitrile (CHETB; 0.040 ± 0.019 µmol/g FW), and allyl isothiocyanate (0.007 ± 0.002 µmol/g FW). Further glucosinolate hydrolysis products were present in small amounts as well (Appendix A). Therefore, up to 23 µmol of CETP were ingested in the preliminary experiment.

For the human pilot intervention study, a beverage prepared from *B. carinata* sprouts was used. It contained 3.34 ± 0.07 µmol/g FW CETP, 0.270 ± 0.054 µmol/g FW allyl isothiocyanate, and 0.132 ± 0.008 µmol/g FW 3-butenenitrile. By consuming 15.2 g of this beverage, volunteers ingested 50.8 µmol of CETP.

### 3.2. Identification of the CETP Metabolite

After ingestion of white cabbage sprouts, urine samples were analyzed using LC-MS. The chromatograms were searched for potential metabolites by extracting the exact mass of potential mercapturic acid pathway metabolites. Two compounds were identified as potential metabolites, as the control urine sample was devoid of them: one CETP-NAC adduct candidate (C_9_H_14_N_2_O_3_S_2_; *m*/*z* [M + H]^+^_calculated_ = 263.05186; *m*/*z* [M + H]^+^_detected_ = 263.05137; δ = 1.86 ppm) and a methylated CETP-NAC adduct candidate (C_10_H_16_N_2_O_3_S_2_; *m*/*z* [M+H]^+^_calculated_ = 277.06751; *m*/*z* [M + H]^+^_detected_ = 277.06784; δ = 1.19 ppm) were identified at 8.9 and 7.3 min, respectively. Below, these two candidates are for simplicity referred to as the *m*/*z* 263 candidate and the *m*/*z* 277 candidate, respectively. Other glucosinolate hydrolysis products, such as allyl isothiocyanate present in the white cabbage sprouts, can be metabolized to isobaric compounds. As a result, allyl isothiocyanate-NAC (C_9_H_14_N_2_O_3_S_2_; *m*/*z* [M + H]^+^_calculated_ = 263.05186) or CETB-NAC (C_10_H_16_N_2_O_3_S_2_; *m*/*z* [M + H]^+^_calculated_ = 277.06751) could also be present in the urine sample. Therefore, these metabolites were synthesized and analyzed by UHPLC-QToF-MS. Thereby, the *m*/*z* 263 candidate metabolite was identified as allyl isothiocyanate-NAC. The remaining *m*/*z* 277 candidate was neither a CETP-*N*-acetyl-l-cysteine methyl ester adduct nor a CETB-NAC adduct, as they had differing retention times.

Due to the fragmentation pattern of the *m*/*z* 277 candidate, it was hypothesized that the sulfur of the former epithio group was methylated and that the metabolite could be *N*-acetyl-*S*-(3-cyano-2-(methylsulfanyl)propyl-cysteine. This compound was synthetized and the identity confirmed by NMR spectroscopy. ^1^H and ^13^C NMR spectra are provided in the Appendix A. When investigating its chromatographic and fragmentation behavior, it was confirmed that *N*-acetyl-*S*-(3-cyano-2-(methylsulfanyl)propyl-cysteine is an urinary metabolite of CETP. As no other metabolite of the mercapturic pathway was detectable, it is suspected that this metabolite is the main one. The structure and fragmentation are provided in Figure 2.

### 3.3. Excretion Kinetics of CETP in Three Volunteers

After ingestion of 50.8 µmol of CETP by drinking the *B*. *carinata* sprout beverage, spot urine was sampled and the concentration of *N*-acetyl-*S*-(3-cyano-2-(methylsulfanyl)propyl-cysteine was analyzed by UHPLC-QToF-MS. In the urine of all three volunteers, the highest CETP-metabolite concentrations were found after 3 h and declined thereafter (Figure 3). In those urine samples, the concentrations of *N*-acetyl-*S*-(3-cyano-2-(methylsulfanyl)propyl-cysteine ranged from 23.9 to 37.2 µM. After 24 h the metabolite nearly reached the pre-intervention level (Figure 3).

## 4. Discussion

Epithionitriles are important hydrolysis products from glucosinolates and many *Brassica* vegetables especially release epithionitriles due to their high abundance of alkenyl glucosinolates ans ESP presence. Cabbage usually releases CETP as the main glucosinolate hydrolysis product [1,25]. However, the metabolism of these compounds in humans is not comprehensively solved. In the present study, the main CETP metabolite, *N*-acetyl-*S*-(3-cyano-2-(methylsulfanyl)propyl-cysteine, which is a mercapturic acid with a methylation at the former epithio-sulfur atom, was identified. In rat urine, similar compounds have been tentatively identified as main metabolites resulting from CETP and CETB [22,23]. However, as those metabolites have been identified using gas chromatography after a derivatization step with diazomethane, it could not be determined if these mercapturic acids were methylated or not [22,23]. Therefore, those metabolites could have been also *N*-acetyl-*S*-(3-cyano-2-(methylsulfanyl)propyl-cysteine. The methylation of the free thiol formed from the epithio-sulfur is not extraordinary: the detoxification of xenobiotic thiols by *S*-methylation is common in mammals, resulting in less reactive methylsulfanyl derivatives [26,27,28]. As the novel metabolite identified here is a mercapturic acid, it is likely formed via the mercapturic acid pathway, similar to the metabolism of isothiocyanates [29]. Initially, conjugation with GSH occurs as proposed in the literature [19,22,23], probably catalyzed by cytosolic glutathione-*S*-transferases that are highly expressed in the liver [30]. Then, the GSH-adduct is further transformed mainly in the kidney into the mercapturic acid, first by renal γ-glutamyltransferase to the *S*-cysteinylglycine conjugate, and then renal cysteinyl-glycine dipeptidase and aminopeptidase M release the *S*-cysteine conjugate, being finally *N*-acylated by renal or hepatic *N*-acetyltransferases to the mercapturic acid [30,31,32]. As many tissues have *S*-methylation activity, the methylation of the sulfur atom might also occur at the GSH-adduct before the final transformations [26]. The mercapturic acid can also be deacylated releasing again the cysteine conjugate. These can be substrates to cysteine conjugate *β*-lyases in the kidney, which can result in further bioactivation [31,33]. It is likely that this mechanism is responsible for the epithionitrile-induced nephrotoxicity observed in rodents, as inhibition of renal cysteine conjugate *β*-lyase by aminooxyacetic acid protected rats from CETB-induced nephrotoxicity [20].

With regard to the concentrations of the CETP metabolite *N*-acetyl-*S*-(3-cyano-2-(methylsulfanyl)propyl-cysteine quantified in the urine of the three volunteers after drinking the *B*. *carinata* beverage, the highest concentrations of *N*-acetyl-*S*-(3-cyano-2-(methylsulfanyl)propyl-cysteine were detected after 3 h. This suggests a very rapid absorption and metabolism of CETP, as was also observed in rats by Broker et al. [22]. In the 3 h urine samples, 30.8 ± 6.7 µM of *N*-acetyl-*S*-(3-cyano-2-(methylsulfanyl)propyl-cysteine were present in average. Thus, the bioavailability of CETP seems to be as fast as that of isothiocyanates: After consumption of Indian cress containing 1000 µmol benzyl glucosinolate, benzyl isothiocyanate-NAC was the main urine metabolite with 98% and its concentration ranged from 830 to 2063 µM between 4 h and 6 h after cress consumption [34]. As in the present study, the CETP amount given was comparatively low, and the cysteine adduct was not found. In the future, further CETP metabolites should be identified in order to study the bioavailability and pharmacokinetics of CETP in more detail.

From this preliminary data, it can be concluded that epithionitriles as main glucosinolate hydrolysis products are rapidly metabolized, forming the methylsulfanyl metabolite *N*-acetyl-*S*-(3-cyano-2-(methylsulfanyl)propyl-cysteine, as identified in human urine. The metabolism seems to be fast, as the highest concentrations were detected 3 h after consumption of a CETP-rich *B. carinata* beverage of smoothie-like consistency. Future investigations are needed to investigate the bioavailability of CETP and to identify the intermediary metabolites in order to evaluate the pharmacokinetics of these compounds.

## Figures and Tables

**Figure 1 nutrients-11-00908-f001:**
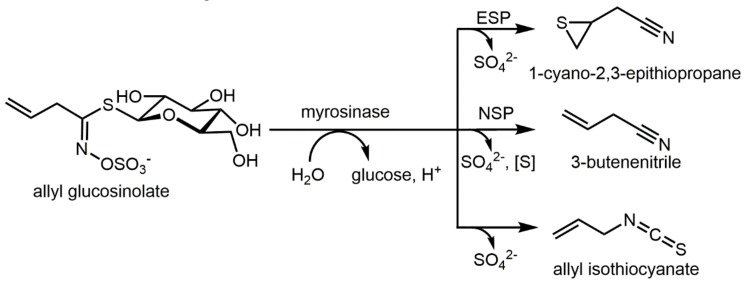
Enzymatic hydrolysis of allyl glucosinolate catalyzed by myrosinase and formation of 1-cyano-2,3-epithionitrile, 3-butenenitrile, and allyl isothiocyanate. ESP: epithiospecifier protein, NSP: nitrile specifier protein. Modified after [10].

**Figure 2 nutrients-11-00908-f002:**
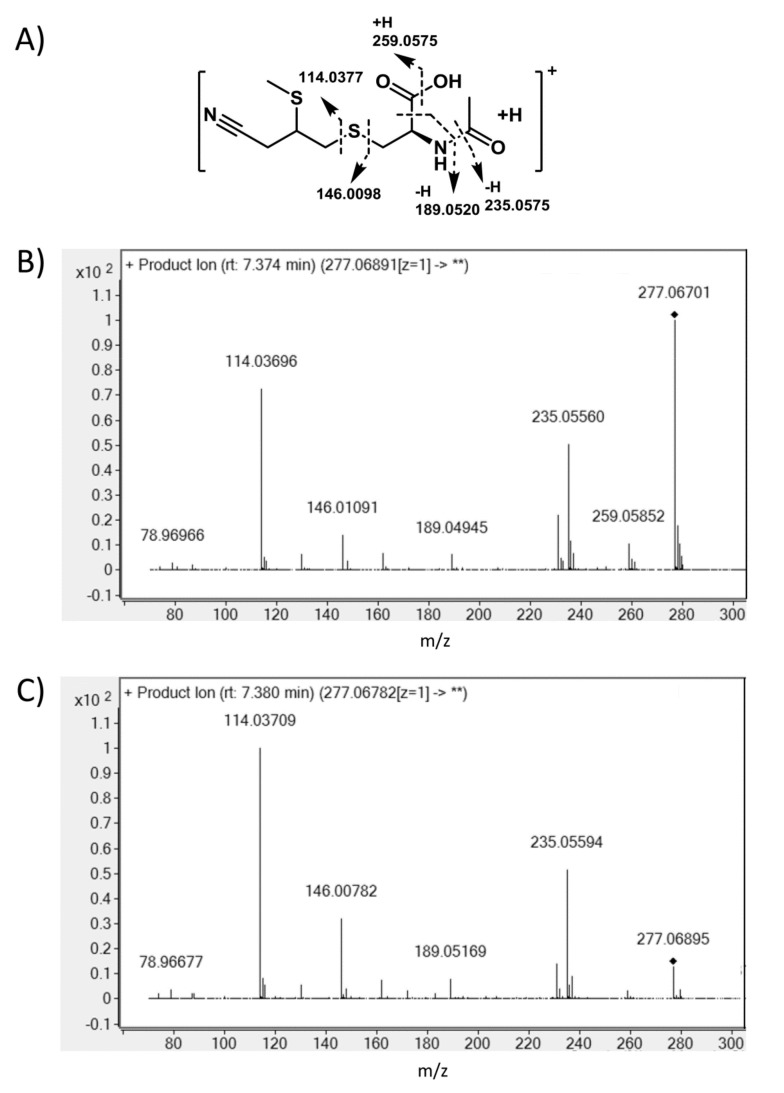
**Liquid chromatography tandem mass spectrometry** (LC-MS/MS) fragmentation and structure of *N*-acetyl-*S*-(3-cyano-2-(methylsulfanyl)propyl-cysteine. (**A**) Structure and fragmentation, (**B**) MS/MS fragments of *m*/*z* 277.06891 with a collision energy of 5 V, and (**C**) MS/MS fragments of *m*/*z* 277.06891 with a collision energy of 10 V.

**Figure 3 nutrients-11-00908-f003:**
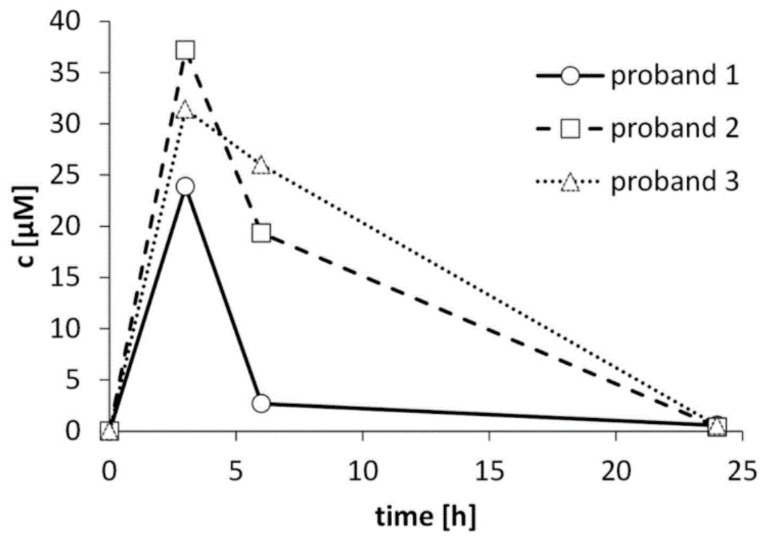
Concentration of *N*-acetyl-*S*-(3-cyano-2-(methylsulfanyl)propyl-cysteine in human urine samples after ingestion of a 1-cyano-2,3-epithiopropane rich *Brassica carinata* smoothie.

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
