# Peer review of "Identification of N-Acetyl-S-(3-Cyano-2-(Methylsulfanyl)Propyl-Cysteine as a Major Human Urine Metabolite from the Epithionitrile 1-Cyano-2,3-Epithiopropane, the Main Glucosinolate Hydrolysis Product from Cabbage"

_nutrients, 2019, doi:10.3390/nu11040908_

Round 1

Reviewer 1 Report

This is a relevant, well-written, interesting and timely presentation of the identification of an apparently major detox product of a major cabbage glucosinolate product. I warmly welcome this paper. The only weakness is in the conclusion that the identified metabolite is the main urinary metabolite of the ingested CETP. The arguments (line 188-190) for this statement are unclear and insufficient. It should be specified how the metabolite was judged to be major (e.g. assuming similar detector response of all metabolites?).

Some sort of preliminary calculation of the molar yield of the CETP detox product from CETP should be presented, even if there is much uncertainty, by estimating the volume of urine collected at 3 h and multiplying with concentration. If this sort of information is impossible to get, please soften the statement of the identified conjugate as the major metabolite.

Minor comments

Title: N and S should be italized in the chemical name

Please consider a simpler title, such as: Identification of a major human metabolite, a mercapturate, of the major cabbage glucosinolate hydrolysis product, an epithionitrile.

A simpler title may give more reader attention, which is what you hope for...

GLS abbreviation is introduced late (line 49), and not explained. By the way, have you noticed that if you instead use GSL or GL, you can put the abbreviation in pluralis by adding an s, while GLS is awkward in this respect.  Later, the abbreviation is not used, and you might want to not abbreviate “glucosinolate” at all.

Line 48. The statement is too strong. Modify, e.g.: In the past decades, epithionitriles and other atypical products from glucosinolates have received intense attention in the pure plant biochemistry field (you can cite a couple of relevant papers by author groups including Wittstock, Kissen, Burow, etc). However, in food and nutritional science, epithionitriles have been nearly neglected…..

Fig 1 sulfate and H+ missing as general products, S missing as coproduct with nitrile. Yes nitriles can be induced by low pH and or Fe2+, but it is not clear cut (a mixture of ITC and nitrile will form), and the main factor in cabbage promoting nitriles in cabbage is probably NSP so I suggest modifying this detail. Indicate CETP at relevant structure. Refer to a recent authoritative review e.g. by Wittstock et al.

At arrow, myrosinase is a catalyst, H2O is a substrate, so the catalyst should stay over the arrow while the substrate should be shown as coming in, with general products (glucose etc coming out) from a suitably curved arrow. There is space for all this if you make the figure a bit broader, it takes up a full page width anyway.

line 52 delete "but detailed investigated" (not perfect grammar and not needed, first part of sentence enough) .

Line 54 do you mean toxification or detoxification (GSK is usually associated with detox, if you do mean tox, please make this extra clear in the sentence).

line 63 abbreviate to GC-MS or put hyphen between …"graphy" and "mass".

line 82: the cystein derivative of CETP

suppl. information 1: an approximate (an not a). As the situation is complicated (2 diastereomers), please show the actual 1H NMR spectrum (a scan of the print is fine), always nice to see the actual data and can be a help for future workers. Please indicate at which C the stereoisomeri originates. Please indicate assignment on the actual spectrum and/or indicate multiplicity after the chemical shifts on the structure, to simplify inspection by readers.

line 95-96 supernatant (5 ml) was diluted 2 fold by adding 5 mL of ethanol. (awkward to treat 5 ml as a plural)

line 96 the sample was centrifuged again

line 123 add spaces cysteine methyl ester (because space is used in ester nomenclature)

line 141 this abstain period was a brave contribution to science. Imagine a week without cabbage!!!

line 146: please specify if possible whether subjects typically urinated between 0h and 3h and between 3h and 6h, all this in order to as good as possible qualify a preliminary estimate of excreted amounts.

page 152 or so, would it be useful for future work to know whether the total number of human guinea pigs were 3 or 4, and whether both male and female subjects participated?

154 glucosinolate hydrolysis products?

157-158 Italize headline?

Line 173-174 reporting found and calculated m/z values. I would use a more traditional format, e.g.: (found, m/z = 263.xxxxx, calculated for C9H14N2O2: 263.xxxyy; delta 1.86 ppm).

suggested extra sentence: "Below, these two candidates are for simplicity referred to as the m/z 263 candidate and the m/z 277 candidate, respectively".

Line 179-180 present version difficult to read because one should pay attention to a long formula and m/z number. Suggested alternative: Thereby the m/z 263 candidate metabolite  was identified as the allyl isothiocyanate NAC adduct.

line 184 can this be written more simple, e.g.: of the m/z 263 candidate metabolite, it was…

line 185, and that the metabolite could be N-acetyl…..……

line 188 I cannot follow this argument. How can chromatographic and fragmentation behavior tell that it is a dominant metabolite? I would say that a standard curve should be prepared and the metabolite quantitated. Alternatively, you should argue in terms of peak areas, assuming same ionization efficiency for all metabolites.

This bit needs some different arguments. For a good paper, it is not absolutely needed that you can conclude this to be the dominating metabolite, you could also go with “would seem to be a dominant or the dominant”, or you could do some rough calculation of the total amount of the metabolite in the delivered urine at 3 h.

line 225 different font

line 245 reconsider conclusion, add modifying statement if relevant

line 247 avoid “very”, fast is enough

Author Response

Comment 1: This is a relevant, well-written, interesting and timely presentation of the identification of an apparently major detox product of a major cabbage glucosinolate product. I warmly welcome this paper. The only weakness is in the conclusion that the identified metabolite is the main urinary metabolite of the ingested CETP. The arguments (line 188-190) for this statement are unclear and insufficient. It should be specified how the metabolite was judged to be major (e.g. assuming similar detector response of all metabolites?).

Reply: This metabolite was the only detectable metabolite in the urine samples. Possible metabolites were identified by searching the chromatograms for the exact masses of the [M+H]+ or [M-H]- ion of potential metabolites formed by the mercapturic acid pathway [GSH conjugate, S-cysteinylglycine (Cys-Gly) conjugate, S-cysteinyl (Cys) conjugate, NAC conjugate] as well as methylated mercapturic acid pathway metabolites (single and double methylated), compare L115-119). Only this one metabolite was from CETP and as we found no other metabolite from the mercapturic acid pathway, we assume that this one is the main one in the urine.

Nevertheless, we added this information at L191ff and modified the conclusion.

Comment 2: Some sort of preliminary calculation of the molar yield of the CETP detox product from CETP should be presented, even if there is much uncertainty, by estimating the volume of urine collected at 3 h and multiplying with concentration. If this sort of information is impossible to get, please soften the statement of the identified conjugate as the major metabolite.

Reply: As we do not have the information about the volume of the urine, we softened the statement of the identified conjugate as the major metabolite.

Minor comments

Comment 3: Title: N and S should be italized in the chemical name

Reply: Revised as requested.

Comment 4: Please consider a simpler title, such as: Identification of a major human metabolite, a mercapturate, of the major cabbage glucosinolate hydrolysis product, an epithionitrile.

A simpler title may give more reader attention, which is what you hope for...

Reply: We considered changing the title. However we would like to keep the name of the metabolite in it. We modified it slightly, so it now is “Identification of N-acetyl-S-(3-cyano-2-(methylthio)propyl-cysteine as a major human urine metabolite from the epithionitrile 1-cyano-2,3-epithiopropane, the main glucosinolate hydrolysis product from cabbage”

Comment 5: GLS abbreviation is introduced late (line 49), and not explained. By the way, have you noticed that if you instead use GSL or GL, you can put the abbreviation in pluralis by adding an s, while GLS is awkward in this respect.  Later, the abbreviation is not used, and you might want to not abbreviate “glucosinolate” at all.

Reply: We removed the abbreviation.

Comment 6: Line 48. The statement is too strong. Modify, e.g.: In the past decades, epithionitriles and other atypical products from glucosinolates have received intense attention in the pure plant biochemistry field (you can cite a couple of relevant papers by author groups including Wittstock, Kissen, Burow, etc). However, in food and nutritional science, epithionitriles have been nearly neglected…..

Reply: Revised as requested.

Comment 7: Fig 1 sulfate and H+ missing as general products, S missing as coproduct with nitrile. Yes nitriles can be induced by low pH and or Fe2+, but it is not clear cut (a mixture of ITC and nitrile will form), and the main factor in cabbage promoting nitriles in cabbage is probably NSP so I suggest modifying this detail. Indicate CETP at relevant structure. Refer to a recent authoritative review e.g. by Wittstock et al.

At arrow, myrosinase is a catalyst, H2O is a substrate, so the catalyst should stay over the arrow while the substrate should be shown as coming in, with general products (glucose etc coming out) from a suitably curved arrow. There is space for all this if you make the figure a bit broader, it takes up a full page width anyway.

Reply: The figure was modified accordingly.

Comment 8: line 52 delete "but detailed investigated" (not perfect grammar and not needed, first part of sentence enough) .

Reply: Revised as suggested.

Comment 9: Line 54 do you mean toxification or detoxification (GSK is usually associated with detox, if you do mean tox, please make this extra clear in the sentence).

Reply: Indeed, toxification is meant. We made this clearer.

Comment 10: line 63 abbreviate to GC-MS or put hyphen between …"graphy" and "mass".

Reply: Revised as requested.

Comment 11: line 82: the cystein derivative of CETP

Reply: Corrected.

Comment 12: suppl. information 1: an approximate (an not a). As the situation is complicated (2 diastereomers), please show the actual 1H NMR spectrum (a scan of the print is fine), always nice to see the actual data and can be a help for future workers. Please indicate at which C the stereoisomeri originates. Please indicate assignment on the actual spectrum and/or indicate multiplicity after the chemical shifts on the structure, to simplify inspection by readers.

Reply: The stereroisomeric center was marked (with *) and a scan of the 1H and 13 C NMR spectra was added. The multiplicity was indicated with the chemical shifts on the structure.

Comment 13: line 95-96 supernatant (5 ml) was diluted 2 fold by adding 5 mL of ethanol. (awkward to treat 5 ml as a plural)

Reply: Corrected as suggested.

Comment 14: line 96 the sample was centrifuged again

Reply: Revised as requested.

Comment 15: line 123 add spaces cysteine methyl ester (because space is used in ester nomenclature)

Reply: Revised as requested.

line 141 this abstain period was a brave contribution to science. Imagine a week without cabbage!!!

Comment 16: line 146: please specify if possible whether subjects typically urinated between 0h and 3h and between 3h and 6h, all this in order to as good as possible qualify a preliminary estimate of excreted amounts.

Reply: It is likely that the volunteers did not urinate between the sampling times. However, we cannot be sure and thus will not make this estimation. In a planned human intervention trial we will address this issue by collecting all urine fractions and measure the volume.

Comment 17: page 152 or so, would it be useful for future work to know whether the total number of human guinea pigs were 3 or 4, and whether both male and female subjects participated?

Reply: In this study for the first trial a man volunteered, for the second trial three female authors volunteered.

Comment 18: 154 glucosinolate hydrolysis products?

Reply: Thank you, this was modified.

Comment 19: 157-158 Italize headline?

Reply: Revised.

Comment 20: Line 173-174 reporting found and calculated m/z values. I would use a more traditional format, e.g.: (found, m/z = 263.xxxxx, calculated for C9H14N2O2: 263.xxxyy; delta 1.86 ppm).

Reply: We would like to keep the format.

Comment 21: suggested extra sentence: "Below, these two candidates are for simplicity referred to as the m/z 263 candidate and the m/z 277 candidate, respectively".

Reply: Revised as suggested.

Comment 22: Line 179-180 present version difficult to read because one should pay attention to a long formula and m/z number. Suggested alternative: Thereby the m/z 263 candidate metabolite  was identified as the allyl isothiocyanate NAC adduct.

Reply: Revised as suggested.

Comment 23: line 184 can this be written more simple, e.g.: of the m/z 263 candidate metabolite, it was…

Reply: Revised as suggested.

Comment 24: line 185, and that the metabolite could be N-acetyl…..……

Reply: Revised as suggested.

Comment 25: line 188 I cannot follow this argument. How can chromatographic and fragmentation behavior tell that it is a dominant metabolite? I would say that a standard curve should be prepared and the metabolite quantitated. Alternatively, you should argue in terms of peak areas, assuming same ionization efficiency for all metabolites. This bit needs some different arguments. For a good paper, it is not absolutely needed that you can conclude this to be the dominating metabolite, you could also go with “would seem to be a dominant or the dominant”, or you could do some rough calculation of the total amount of the metabolite in the delivered urine at 3 h.

Reply: As already answered to Comment 1 no other metabolite was detectable (although searched for). As stated above this information was added. The statement was softened.

Comment 26: line 225 different font

Reply: Modified.

Comment 27: line 245 reconsider conclusion, add modifying statement if relevant

Reply: We would like to leave the conclusion like that.

Comment 28: line 247 avoid “very”, fast is enough

Reply: Revised as suggested. 

Reviewer 2 Report

Another contribution by Hanschen and coll about the metabolization of epithionitriles from cabbages. Using synthesized standards allows confirmation of Van Steenhouse’s results. This might be published in Nutrients after minor revision. Some referee’s comments:

-        In title and throughout the paper (l.19 etc…) including suppl. material: “methylthio” should be replaced by “methylsulfanyl” (updated nomenclature)

-        p1 l.17 “After” (no boldface) / l.21 “Then, in…” / l.24 “consumption” / l.25 “Thus, epithionitriles…” / l.26 “…future, more…” / l.27 “…of this…” / l.35 “…their glucosinolate precursors…” / l.39 “Savoy cabbage” – the sentence “Nevertheless…amounts [1, 4-6]” is too long, please rephrase.

-        p2, l.48 though it is likely that epithionitriles have been “nearly neglected” in the last decades, the studies by Palmieri’s group in the nineties should be mentioned – please search literature. / l.52 “…investigated in detail.” / l.54 what is “epithionitrile toxification in rats” ? / l.70 “…samples, as part of a human…” / l.74 “…cysteine methyl ester…” / l.82 “N-acetyl-S-(3-cyano-2-(methylsulfanyl)propyl

-        p3, l.96 “…was centrifuged again…” / l.117 why 4 significant figures in the first m/z and 5 in the other ?

-        p4, l.138 “Human intervention…” / ll.157-158 this (no verb appearing) sounds like a title – is that the case ? please check

-        p5, l.182 “…cysteine methyl ester…” / ll.186 & 189 “…methylsulfanyl…”

-        p6, (throughout) “…methylsulfanyl…” / ll. 217-218 “…S-methylation…derivatives” should be rephrased / l.223 “…into the mercapturic acid” / ll. 224-225 check font

-        p7 (throughout) “…methylsulfanyl…” / l.239 “…a fast as that of…” / l.246 “…the methyl sulfide metabolite…” or “methylsulfanyl metabolite…” / l.249 “consistency”

-        p7 & p8 why not use abbreviated journal names (cf instructions for authors) ?

-        p8, ll.279-280 “…EPIC-Heidelberg…”

Author Response

Another contribution by Hanschen and coll about the metabolization of epithionitriles from cabbages. Using synthesized standards allows confirmation of Van Steenhouse’s results. This might be published in Nutrients after minor revision. Some referee’s comments:

Comment 1: -        In title and throughout the paper (l.19 etc…) including suppl. material: “methylthio” should be replaced by “methylsulfanyl” (updated nomenclature)

Reply: Revised as suggested.

Comment 2: -        p1 l.17 “After” (no boldface) / l.21 “Then, in…” / l.24 “consumption” / l.25 “Thus, epithionitriles…” / l.26 “…future, more…” / l.27 “…of this…” / l.35 “…their glucosinolate precursors…” / l.39 “Savoy cabbage” – the sentence “Nevertheless…amounts [1, 4-6]” is too long, please rephrase.

Reply: Revised as requested.

Comment 3: -        p2, l.48 though it is likely that epithionitriles have been “nearly neglected” in the last decades, the studies by Palmieri’s group in the nineties should be mentioned – please search literature. /

Reply: Web of Science was searched for the author name “Palmieri” and refined with “glucosinolate”. Several publications of Sandro Palmieris group were found. However none of ot was about epithionitriles. There were some papers working with alkenyl glucosinolates (such as epiprogoitrin) but activation was only with myrosinase (no ESP, so epithionitriles seemed to be not investigated). In one paper they isolated 2-hydroxy-3-butenyl cyanide (DOI: 10.1016/S0040-4039(00)61525-3 ) and in the other (5R)-5-vinyloxazolidme-2-thione (https://doi.org/10.1016/0957-4166(94)80145-2), but they did not form the epithionitrile. Also google scholar search was not successful in this respect. If you could give us a title or DOI, we would be happy to include the publication.

Nevertheless, the text was rephrased: “In the past decades, epithionitriles and other atypical products from glucosinolates have received intense attention in the pure plant biochemistry field (Burow, Markert, Gershenzon, & Wittstock, 2006; Kissen, Hyldbakk, Wang, Sørmo, Rossiter, & Bones, 2012; Wittstock & Burow, 2010; Zhang, Wang, Liu, Xie, Wang, Mu, et al., 2016). However, in food and nutritional science, epithionitriles have been nearly neglected and here glucosinolate research mainly focussed on isothiocyanates, another important class of glucosinolate hydrolysis products that are valued especially for their potential cancer preventing effects [10].”

Comment 4: l.52 “…investigated in detail.”

Reply: This phrase deleted due to the suggestion of another reviewer.

Comment 5: l.54 what is “epithionitrile toxification in rats” ?

Reply: Toxification means the metabolic conversion of a substance into a toxin. The sentence was slightly altered to “Conjugation with glutathione (GSH) is thought to be the initial step in epithionitrile metabolization but also toxification in rats [14]…”

Comment 6: l.70 “…samples, as part of a human…” / l.74 “…cysteine methyl ester…” / l.82 “N-acetyl-S-(3-cyano-2-(methylsulfanyl)propyl

Reply: Revised as suggested.

Comment 6: -        p3, l.96 “…was centrifuged again…” / l.117 why 4 significant figures in the first m/z and 5 in the other ?

Reply: Revised.

Comment 7: -        p4, l.138 “Human intervention…” / ll.157-158 this (no verb appearing) sounds like a title – is that the case ? please check

Reply: Checked and corrected.

Comment 8: -        p5, l.182 “…cysteine methyl ester…” / ll.186 & 189 “…methylsulfanyl…”

Reply: Revised

Comment 9: -        p6, (throughout) “…methylsulfanyl…” / ll. 217-218 “…S-methylation…derivatives” should be rephrased / l.223 “…into the mercapturic acid” / ll. 224-225 check font

Reply: Revised as suggested.

Comment 10: -        p7 (throughout) “…methylsulfanyl…” / l.239 “…a fast as that of…” / l.246 “…the methyl sulfide metabolite…” or “methylsulfanyl metabolite…” / l.249 “consistency”

Reply: Revised.

Comment 11: -        p7 & p8 why not use abbreviated journal names (cf instructions for authors) ?

Reply: Revised as suggested.

Comment 12: -        p8, ll.279-280 “…EPIC-Heidelberg…”

Reply: Revised as suggested.